# Evaluation of Cooked Rice for Eating Quality and Its Components in *Geng* Rice

**DOI:** 10.3390/foods12173267

**Published:** 2023-08-30

**Authors:** Cui Li, Shujun Yao, Bo Song, Lei Zhao, Bingzhu Hou, Yong Zhang, Fan Zhang, Xiaoquan Qi

**Affiliations:** 1Key Laboratory of Plant Molecular Physiology, Institute of Botany, Chinese Academy of Sciences, Nanxincun 20, Fragrant Hill, Beijing 100093, China; licui@ibcas.ac.cn (C.L.); yaoshujun@ibcas.ac.cn (S.Y.); bosong@ibcas.ac.cn (B.S.); houbz@ibcas.ac.cn (B.H.); zhangf32@ibcas.ac.cn (F.Z.); 2University of Chinese Academy of Sciences, Yuquan Road 19, Beijing 100049, China; 3China National Botanical Garden, Nanxincun 20, Fragrant Hill, Beijing 100093, China; 4Tonghua Academy of Agricultural Sciences, Hailong Town, Meihekou 135007, China; nky5158@163.com; 5LUSTER LightTech Co., Ltd., Yard No.13, Cuihu Nanhuan Road, Beijing 100094, China; yongzhang@lusterinc.com

**Keywords:** appearance, predicted model, rice eating quality, viscosity

## Abstract

At present, ‘‘eating well” is increasingly desired by people instead of merely ‘‘being full”. Rice provides the majority of daily caloric needs for half of the global human population. However, eating quality is difficult to objectively evaluate in rice breeding programs. This study was carried out to objectively quantify and predict eating quality in *Geng* rice. First, eating quality and its components were identified by trained panels. Analysis of variance and broad-sense heritability showed that variation among varieties was significant for all traits except hardness. Among them, viscosity, taste, and appearance were significantly correlated with eating quality. We established an image acquisition and processing system to quantify cooked rice appearance and optimized the process of measuring cooked rice viscosity with a texture analyzer. The results show that yellow areas of the images were significantly correlated with appearance, and adhesiveness was significantly correlated with viscosity. Based on these results, multiple regression analysis was used to predict eating quality: eating quality = 0.37 × adhesiveness − 0.71 × yellow area + 0.89 × taste − 0.34, *R*^2^ = 0.85. The correlation coefficient between the predicted and actual values was 0.86. We anticipate that this predictive model will be useful in future breeding programs for high-eating-quality rice.

## 1. Introduction

Rice (*Oryza sativa* L.) is the most important food crop in the world, especially in Asia, because it is a staple food for more than half of the world’s population [1]. For many years, the goals of rice breeding programs have mainly focused on yield and its related traits, but it has proven increasingly difficult to meet the demands of consumers and the marketplace for rice quality [2,3,4].

The main constituents of the rice grain are starch (70–80%), protein (7–10%), lipids (<1%), trace elements, and small amounts of various other compounds [5]. Rice grain quality is a combination of nutritional quality, storage quality, processing and appearance quality (commercial quality), and cooked rice eating quality [6]. There are clear national standards for commercial quality in China, but not for eating quality, so even some first-class rice varieties may have poor eating quality. For the improvement of rice eating quality via breeding, however, there have been few studies on the genetic basis of eating quality and its component traits, mainly due to the lack of effective evaluation methods [1,7,8,9,10]. 

At present, sensory tests conducted with trained panels are the most appropriate way to evaluate eating quality and its components, involving Integrated Values of Organic Evaluation (IVOE, eating quality), appearance, hardness, viscosity, taste, and fragrance. This method not only requires a large number of samples and high-generation homozygous rice lines, but is also relatively inefficient, with high labor costs and poor repeatability [11]. Therefore, there is an urgent need to establish an objective, instrument-based evaluation method [12].

The most common instrument used to measure the hardness and viscosity of cooked rice kernels is the texture analyzer [13,14,15,16,17,18]. It obtains the force–displacement curve by a double compression test, typically with two rice kernels, which is less reliable and accurate than a test performed on bulk samples [12]. Some color-measuring instruments, such as a portable spectrophotometer CM-600D (Konica Minolta Sensing Inc., Osaka, Japan) [19], the HunterLab MiniScan XE Plus Diffuse LAV M072096 colorimeter (Hunter Associates Laboratory, Inc., Reston, VA, USA) [8], and the Chroma Meter CR700d/600d (Konica Minolta Sensing Inc., Osaka, Japan), are occasionally used for cooked rice [20,21]. However, there are always inconsistencies with artificial sensory tests, and large uniform samples are still needed, making these methods unsuitable for objectifying appearance quality [22]. In addition, to improve the prediction accuracy of rice eating quality, some mining techniques based on the measurement of rice components and physicochemical characteristics have been developed using statistical methods, such as principal component analysis (PCA), multiple linear regression analysis (MLRA), and partial least squares regression analysis (PLSRA) [23,24,25]. However, there are no reports that combine these techniques and the components of eating quality to systematically predict eating quality in cooked rice. 

The evaluation of rice eating quality is generally divided into the *Geng* and *Xian* groups, and most consumers in China prefer the former. In this study, we optimized the existing evaluation methods, explored new methods to objectify indicators that are highly correlated with IVOE using the *Geng* rice grown in northeastern China, and established a prediction model using the objective indicators for eating quality. There was a strong correlation between the prediction model and the sensory tests of IVOE. Our work provides an accurate, low-cost, convenient, and rapid method to evaluate rice eating quality for the breeding of high-quality varieties. The prediction model, combined with responsive measuring equipment, will help to build a cooked rice taste instrument suitable for evaluating *Geng* rice in China.

## 2. Materials and Methods

### 2.1. Plant Material and Growth Conditions

A total of 322 *Geng* rice varieties were intentionally selected to represent varying eating qualities. Some varieties have been rewarded by the national or regional high eating quality rice variety evaluation (China), such as varieties T9029 and T9169. Other varieties, whose eating qualities range from high to low, have been selected by sensory tests for several continuous years. The trained panels conventionally give scores on the appearance, fragrance, taste, hardness, viscosity, and integrated quality using their senses of sight, smell, taste, and touch. 

The varieties in this study are mainly from Jilin province, Heilongjiang province in China, and partially from Japan, South Korea, North Korea, and other provinces of China, as listed in Appendix A. The rice was grown during the normal rice-growing seasons at the experimental station in Tonghua (125° E, 42°32′ N), Jilin Province, China. The major environmental conditions were a mean temperature of 25–30 °C, and mean precipitation of 870 mm during the rice flowering. All seeds were planted in a seedbed in mid-April and transplanted to the field in mid-May. The planting density was 16.5 cm between plants in a row, and the rows were 26 cm apart. The plants were grown in a randomized complete-block design with two replicates in both 2020 and 2021. Field management, including irrigation, fertilizer application, and pest control measures, essentially followed normal agricultural practices for growing rice. Mature seeds were randomly collected and pooled for sensory evaluation and other measurements.

### 2.2. Sensory Testing Using a Taste Panel

Milled rice (moisture content ranged from 13.5% to 14.5%) was cooked according to the protocol “GB/T 15682-2008 method for sensory test of paddy or rice cooking and eating quality”. Dry-milled head rice (300 g) was rinsed two times and soaked for 10 min in distilled water. The rice was cooked using an electric rice cooker, and the ratio of rice/water was 1:1.35 *w*/*w*. After finishing the automatic cooking cycle, the cooked rice was kept in the cooker for 30 min. Samples were transferred to plates and kept at room temperature for about 10 min until they had cooled to 35–37 °C. Eleven well-trained panel members had participated in the Training and Assessment of Rice Eating Quality Evaluators organized by the Panjin Northern Rice Production Association (Liaoning, China). Six females and five males were from different provinces of China and worked at the Tonghua Academy of Agricultural Sciences. The eating quality was assessed based on the appearance, fragrance, taste, hardness, viscosity, and integrated values of organic evaluation (IVOE) from −3 to +3 compared to a rice reference variety, Qiutianxiaoting (value = 0). For each trait, seven scores, including −3, −2, −1, 0, +1, +2, and +3 were designed, representing the worst, worse, bad, the same, good, better, and best, respectively [9]. The sensory test value of each variety with three replications was the average of the values determined by 11 panel members.

### 2.3. Construction of an Image Acquisition System

We constructed an image acquisition system, which cooperated with Luster. The system consisted of a telecentric lens and an LED lamp (Philips (China) Investment Co., Ltd., Shanghai, China) with a color rendering index of >90% and a color temperature of 6500 K, gray acrylic material, and a plastic container. The 30 cm diameter LED dome lamp was placed on top of the device, and the lens was in the middle of the lamp. The lens was positioned on a stand at 11 cm above the sample, and the lens was connected to a computer. The shell of the image acquisition system was made of a gray acrylic sheet (Appendix A).

### 2.4. Imaging for Cooked Rice

Detailed process of cooked rice imaging: (1) White A4 paper was used to calibrate the white balance of the LED lamp, which was used to ensure that the samples were colored with the same light source. (2) The volume of rice was equal to the mark scale of the container, nearly 30 cm^3^. It is required that the rice cannot be pressed forcefully, and there was no large gap among the rice grains. (3) A computer was connected to the lens, and pylon Viewer 6.0.13.7126 was used for imaging (exposure value per microsecond was 7000) (Appendix A). (4) Images were saved in the TIFF format and with a resolution of 8 megapixels (3264 × 2448). To ensure the equality of the lighting, all images were taken half an hour after the lamp was switched on. Each sample was imaged for four times independently.

### 2.5. Extracting the Yellow Areas from the Images

Image analysis was performed as follows: (1) The original rice image was imported into ImageJ, and the chromatogram was examined to find the appropriate spectrum for extracting the rice yellow area. (2) Red, green, and blue (RGB) pictures were imported into MATLAB9.9. (3) Command rgb2hsv was used to convert RGB into hue, saturation, and value (HSV) format. (4) The imbinarize function was used to binarize the adjusted image by hue score (H < 0.167 is black). (6) The color of the binarized image was inverted, and the image will have the yellow region as 255 (white) and the others were 0 (black). White pixels were extracted and colored with yellow. (7) The final step was to calculate the total number of pixels in the yellow region by using the function “find()” to extract the pixels equal to 0, and using the function “length()” to count the number of pixels that were extracted, which gave the total area in the image (Appendix A).The above script file was specifically analyzed using MATLAB 9.9.

### 2.6. Texture Measurement

Dry-milled head rice (10 g) was rinsed twice and soaked for 10 min in distilled water in a 30 mL Zisha cup; there were four replicates for each sample. The rice was cooked using a steamer for 20 min with a rice/water ratio of 1:1.35 *w*/*w*, after which it was kept it warm for 20 min. The cooked rice was taken out and covered with a paper cup at room temperature for about 1.5 h (Appendix A). Analysis of the textural attributes was performed on a texture analyzer (Bossin Tech, Shanghai, China) using a 36 mm cylinder probe. The texture analyzer settings were as follows: The pre-test speed, test speed, and post-test speed of the plunger were set to 2.0, 1.0, and 2 mm/s, respectively. The compression distance was 35% strain, and the (auto) trigger force was 25 gf. In order to avoid differences resulting from the different batches, the parameters measured by the texture analyzer were determined by comparing the samples with the control.

### 2.7. Statistical Analyses

Correlation analysis was analyzed with SPSS (version 16.0; SPSS Inc, Chicago, IL, USA). The paired sample *t*-test was conducted for comparison purposes between the instrument-derived values and the sensory evaluation with a 95% confidence interval.

Broad-sense heritability (*h*^2^) was calculated using the following equation by treating each rice accession as a random effect and the biological replication as a replication effect using one-way ANOVA: *h*^2^ = var _(G)_/(var _(G)_ + var _(E)_), where var _(G)_ and var _(E)_ were the variances derived from the genetic and environmental effects, respectively.

The best subset regression is a model selection approach that consists of testing all possible combinations of the predictor variables, and then selecting the best model based on some statistical criteria. The R function “regsubsets ()” in the “leaps” package could be used to identify the different best models for the different indicators.

## 3. Results

### 3.1. Characteristics of the Major Components of Eating Quality of Cooked Rice

A total of 322 *Geng* rice varieties were intentionally selected to represent varying eating qualities. We used a relatively unified artificial sensory test to identify the IVOE and its components, such as the hardness, viscosity, appearance, taste, and fragrance of the cooked rice. The frequency of the cooked rice IVOE and its components showed normal distributions (Figure 1). The values were in the ranges of −2.50 to 0.40, −2.12 to 1.00, −2.06 to 0.49, −2.56 to 0.50, −0.93 to 1.12, and −1.31 to 0.58 for the IVOE, viscosity, appearance, taste, hardness, and fragrance, respectively. The average values were −0.82, −0.56, −0.49, −0.66, 0.06, and −0.20 for the IVOE, viscosity, appearance, taste, hardness, and fragrance, respectively (Figure 1), indicating the samples selected in this study with different eating qualities, including high, medium, and low values for each trait.

The *F*-test of the variation in IVOE and its component traits showed that there were significant differences among the varieties for all phenotypes except hardness, and hardness was greatly influenced by the environment (Table 1).

The eating quality of rice is jointly influenced by genetics and the environment. In order to avoid the impact of environmental factors on the eating quality, we conducted experiments in two years. The trait with a high correlation coefficient between the two years indicates the reliability of the experimental results. In a comparison of the two crop years, the IVOE, viscosity, appearance, taste, and fragrance were significantly correlated to themselves, and the *r* values were 0.50, 0.45, 0.24, 0.36, and 0.23 (*p* < 0.01), respectively. However, hardness was greatly influenced by the environment, and the correlation coefficient between the two years was 0.04 (Figure 2), which is consistent with the results of the ANOVA. Broad-sense heritability refers to the percentage of the total genetic variance to the total variance. The numerical range of broad-sense heritability is 0 to 1, and a higher value indicates a greater contribution of genetic factors in individual trait variation. The results for the broad-sense heritability showed that IVOE (*h*^2^ = 0.78), viscosity (*h*^2^ = 0.76), and taste (*h*^2^ = 0.71) had high heritability, while appearance (*h*^2^ = 0.5) and fragrance (*h*^2^ = 0.38) had medium heritability (Figure 3A). Combining the results of the ANOVA and broad-sense heritability analysis showed that the variation among the rice varieties in IVOE and its component traits was due to inheritance, although the heritability differed for the individual traits, except for hardness.

### 3.2. Relationships between IVOE and Its Component Traits

Correlation analysis showed that the IVOE was significantly positively correlated with viscosity (*r* = 0.86), taste (*r* = 0.84), and appearance (*r* = 0.80), respectively. The IVOE was significantly negatively correlated with hardness, with a correlation coefficient of −0.31. Varying degrees of correlation were found between any two traits among viscosity, taste, appearance, hardness, and fragrance; the strongest correlation was between taste and appearance (*r* = 0.82) and the lowest correlation was between appearance and fragrance (*r* = −0.16). Therefore, viscosity, appearance, and taste are the principal component traits of IVOE, and they were selected for further objectification (Figure 3B).

### 3.3. Evaluation of the Major Component Traits for IVOE

#### 3.3.1. The Yellow Area Objectively Indicates Cooked Rice Appearance

Researchers have employed image processing in many fields; however, there are almost no reports of using this method to analyze the appearance of cooked rice. Before the experiment, an image acquisition system was constructed to standardize the procedure of acquiring images. Six samples with high and low appearance values were selected to test the system. The average appearance values for the cooked rice samples shown in Figure 4A–F were −2.06, −2.0, −2.0, 0.75, 0.40, and 0.39 for T9374, T9329, T9080, T9027, T9135, and T9028, respectively. By observing the images, we found that cooked rice samples with high appearance values tended to be white and glossy, while cooked rice samples with low values tended to be yellow with weak gloss, which is consistent with the evaluation results for cooked rice appearance determined by the sensory tests. Therefore, images taken by our system accurately reflect the features of cooked rice appearance.

We extracted different types of feature values from the images for the quantitative analysis of the cooked appearance features, such as the color attributes and the yellow areas. The color models CIEL*a*b*and hue–saturation–intensity (HSI) were available to extract the color attributes, which were defined and used as international standards. In the CIEL*a*b* color model, L is a measure of the brightness from black (0) to white (100). Parameter (a) describes red–green colors, with +a values indicating redness and −a values indicating greenness. Parameter (b) describes yellow–blue colors, with +b values indicating yellowness and -b values indicating blueness [21]. However, no parameters of the color models CIEL*a*b* and HSI were significantly correlated with the cooked rice appearance (Appendix A). The yellow areas of the images were the percentage of yellow pixels in the total number of pixels. The images with small yellow areas have high appearance values. In contrast, the yellow areas were large for cooked rice with low appearance values. For example, the appearance value for T9028 was 0.39, and the yellow area was 7% (Figure 4H). In contrast, the appearance value for T9080 was −2.00, and the yellow area of the image was 53% (Figure 4G). We analyzed the yellow areas of all samples grown in two consecutive years and found that the yellow areas from the images ranged from 5.78% to 62.55% and from 5.00% to 60.13% in 2020 and 2021, respectively. Correlation analysis showed that the sizes of the yellow areas were significantly negatively correlated with the appearance value, and the respective correlation coefficients were −0.40 and −0.37 (Figure 5).

The above results show that the image analysis system constructed in this study can accurately capture features of the appearance of cooked rice. In the future, yellow areas extracted from the images can be used as objective indicators to evaluate cooked rice appearance.

#### 3.3.2. Adhesiveness Objectively Indicates Cooked Rice Viscosity

A texture analyzer has been used to analyze cooked rice texture with some success, but there are still some steps that are worth optimizing to better explain the texture attributes of cooked rice. In this study, we optimized the process of measuring the viscosity of cooked rice using a texture analyzer. For example, we used a Zisha cup instead of a metal vessel to hold the rice, and the original state bulk samples were used to measure the parameters. For cooked rice samples with high viscosity values, the area of A3 (adhesiveness) was large, while it was small for the samples with low viscosity values. To illustrate this, the viscosity value for T9051 was 1.00, and the area of A3 was 1366.48 (Figure 6B), while the viscosity value for T9216 was −1.42 and the area of A3 was 300.53 (Figure 6C). The area of A3 was significantly different between T9051 and T9216.

We measured the texture profile analysis (TPA) parameters of more than 300 rice samples (with 4 replicates per variety). The TPA parameters that were significantly correlated with the viscosity value were adhesiveness, springiness, chewiness, and cohesiveness (*p* < 0.01) in 2020 (Appendix A). It is worth noting that the adhesiveness was significantly positively correlated with the viscosity value, with *r* = 0.42 (Figure 7A). We used the same method in 2021 and found that the TPA parameters that were significantly correlated with the viscosity value were adhesiveness, springiness, and gumminess (*p* < 0.01). Among these parameters, adhesiveness was significantly positively correlated with the viscosity value, with *r* = 0.58 (Figure 7B). Both chewiness and resilience were correlated with viscosity at the 0.05 level (Appendix A). Combining the results from 2020 and 2021, adhesiveness was significantly positively correlated with viscosity values from the sensory tests. Therefore, our results show that adhesiveness measured with a texture analyzer can be used as an objective indicator for cooked rice viscosity and can be used for measuring cooked rice viscosity in the future.

#### 3.3.3. Establishment of an Objective Prediction Model for Rice Eating Quality

A multiple regression analysis to estimate the rice eating quality yielded different contributing variables. The variable model was identified as optimal for rice eating quality. According to the *R*^2^ (goodness-of-fit measure), the best performer was the model with three variables from the rice eating quality component traits; they were viscosity, appearance, and taste. The model was IVOE = 0.20 × viscosity + 0.35 × appearance + 0.59 × taste + 0.13, *R*^2^ = 0.87 (Model 1). Using this model, we predicted the IVOE of 78 *Geng* rice varieties. The correlation between the predicted IVOE and the sensory test values was high, with *R*^2^ = 0.89 (Figure 8A).

An attempt was made to use objective indicators as substitutes for the sensory test for viscosity or appearance, or both. These trait components were considered to contribute significantly to the IVOE. We constructed three other evaluation models for IVOE (Models 2–4).

Model 2: IVOE = 0.27 × adhesiveness + 0.40 × appearance + 0.67 × taste − 0.35, *R*^2^ = 0.87;Model 3: IVOE = 0.27 × viscosity − 0.57 × yellow area + 0.76 × taste − 0.05, *R*^2^ = 0.86;Model 4: IVOE = 0.37 × adhesiveness − 0.71 × yellow area + 0.89 × taste − 0.34, *R*^2^ = 0.85.

We used Models 2–4 to predict the IVOE with the 78 same rice varieties used for Model 1 and found that the correlation coefficients between the predicted IVOE and the sensory test values was high, with *R*^2^ values of 0.89 (Model 2), 0.87 (Model 3), and 0.86 (Model 4) (Figure 8B–D). All of the above results demonstrate the availability and accuracy of the objective indicators and predicted models.

## 4. Discussion

Rice is normally consumed in the form of whole grains, rather than special forms, such as rice flour, rice noodles, and so on [12,19,26,27]. Using cooked rice in research is the foundation for studying eating quality in rice. Quantitative trait loci (QTLs) usually refer to the positions of genes that control quantitative traits on a genome. Some QTLs with a large region for cooked rice quality have been mapped in the past two decades [27,28,29,30,31,32,33]. However, few genes in these QTLs have been identified due to the lack of objective methods for the IVOE and its components. Our work has established a foundation for the molecular analysis of rice eating quality, and we selected representative *Geng* rice varieties to effectively illustrate the genetic mechanisms that underlie eating quality.

In this study, the differences in the IVOE and its component traits among varieties, except for hardness, were caused by genetic variation (Table 1), and the IVOE was significantly positively correlated with viscosity, appearance, and taste in cooked rice (Figure 3B). This result is consistent with results of previous studies [1,12,34]. This once again shows that the viscosity, appearance, and taste of cooked rice are the principal component traits of the IVOE and need to be objectified.

To accurately evaluate the appearance characteristics of cooked rice, we constructed an image acquisition and processing system as described below. In the image acquisition system, the LED light sources with different color rendering indices (CRI) were used, and it was found that when the CRI > 90, the appearance of cooked rice was restored nearly to the realistic color. The color temperature was 6500 K, which is similar to daylight white [8,35,36]. A telecentric lens was selected to measure objects that were not strictly on the same surface. The shell of the image acquisition system was made of gray acrylic material, which neither absorbs nor reflects light, and is easy to carry and handle. The pictures taken by this system could truly reflect the features of cooked rice appearance, and were consistent with the sensory tests (Figure 4).

To objectively measure the color characteristics of cooked rice, we attempted to use color models such as CIEL*a*b* and HSI. It has been reported that the L value of the CIEL*a*b color model and the I value of the HSI color model are significantly positively correlated with appearance, and that the value of parameter (b) of the CIEL*a*b*color model is significantly negatively correlated with appearance [37,38]. The parameters L*, a*, and b* were generally used to describe the cooked rice appearance with large-scale differences. However, the appearance differences of cooked rice in this study were slight, so no parameters of CIEL*a*b* were significantly correlated with the cooked rice appearance. The differences in both appearance characteristics of rice varieties and degrees of polished rice in grinding process, which result in inconsistent results with other researches in the HSI model. Thus, we developed a new color model for extracting the yellow areas in images of cooked rice. In the independent experiments conducted in 2020 and 2021, the yellow areas were significantly negatively correlated with appearance (Figure 5). We think that the image acquisition system constructed for our study will be a quick and effective tool for determining the appearance characteristics of cooked rice in future research.

Accurate measurements of viscosity are also important for studying the IVOE formation mechanism. We optimized the process of measuring viscosity using a texture analyzer: (1) cooking rice with a Zisha cup avoided the influence of moisture in the container; (2) cooked rice was neither stirred nor pressed into cakes to keep the rice in its original state and to simplify the measurement process; and (3) bulk samples, not individual kernels, were used, which is more reliable and accurate [12]. Using our optimized method, we found that the correlation coefficients between adhesiveness and viscosity were the highest (Figure 7, Appendix A). This result is in line with the common sense of physical mechanics and previous research results [38]. Therefore, we showed that the process we used to objectively measure the viscosity with a texture analyzer was feasible, and that the results are reliable and stable.

Recently, Japanese researchers have developed a taste analyzer that converts various physical and chemical parameters of rice into a “taste” value based on the correlation between near-infrared reflectance measurements of key ingredients and sensory values [39,40]. The taste value assigned by the taste analyzer may be correlated with the IVOE [40,41]. However, there are few reports that use the models of direct indicators (such as appearance, viscosity, and taste) to predict the IVOE. In this study, we used multiple regression analysis to estimate the rice eating quality based on appearance, viscosity, and taste, and Model 1 had a high correlation coefficient (*R*^2^ = 0.87) between the predicted IVOE and the IVOE determined by the sensory tests (Figure 8A). We used objectified indictors to substitute for viscosity or appearance, or both, to construct three other predictive models (Models 2–4). Their correlation coefficients were close to that of Model 1 (Figure 8). To a certain degree, all of the above results show that objectified indicators can be used instead of sensory test values to evaluate the IVOE. Unfortunately, taste is still evaluated by sensory tests rather than objective indicators, which is a top priority for future research.

The results of this study increase our understanding of the genetic basis of eating quality and its component traits in rice and provide new insights for rice breeders to select rice varieties with desirable eating quality for cultivation. In the future, integrating these sub-elements (an imaging acquisition system and a texture analyzer) with prediction software (an eating quality prediction model) has the potential to establish an eating quality instrument, providing us with a faster, more convenient, and more accurate method for evaluating the eating quality of *Geng* rice.

## 5. Conclusions

In this study, we confirmed that viscosity, appearance, and taste are the major contributors to the IVOE, and the differences among rice varieties were caused by inheritance. The viscosity and appearance, indicated by adhesiveness and the yellow areas of images, were successfully objectified using the texture analyzer and a new image processing system. The prediction model was constructed as Model 4: IVOE = 0.37 × adhesiveness − 0.71 × yellow area + 0.89 × taste − 0.34, *R*^2^ = 0.85. Its correlation coefficient with the values determined by the sensory tests was *R*^2^ = 0.86. We hope the prediction model for the IVOE by objective indicators will be used in instruments for evaluating cooked rice eating quality.

## Figures and Tables

**Figure 1 foods-12-03267-f001:**
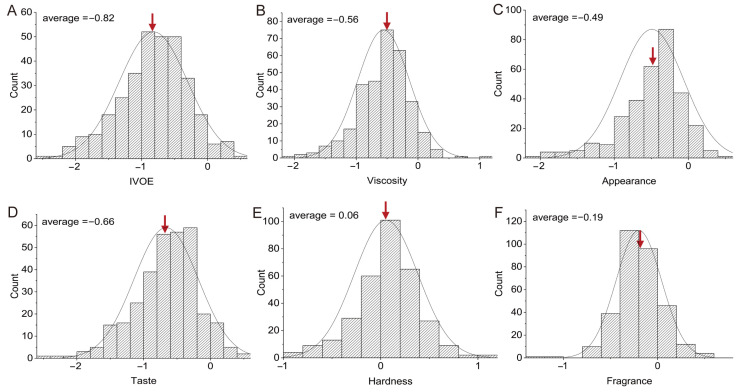
The frequency distributions of cooked rice IVOE and its components. The frequency distributions of cooked rice IVOE (**A**), viscosity (**B**), appearance (**C**), taste (**D**), hardness (**E**), and fragrance (**F**). The arrows show the average values of IVOE and each of its component traits.

**Figure 2 foods-12-03267-f002:**
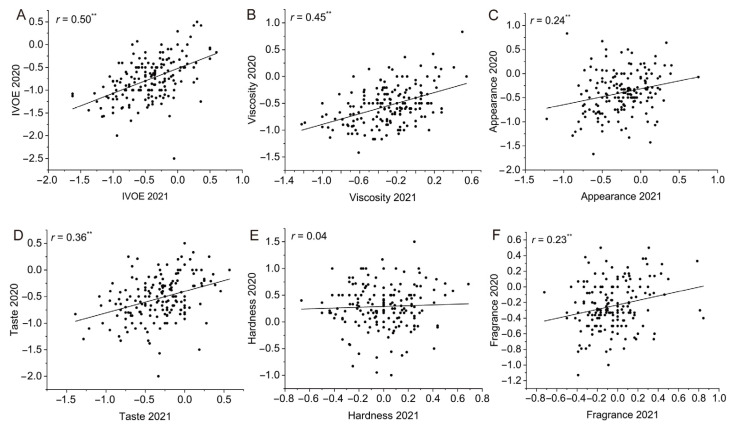
Correlation analysis of IVOE and its component traits for the two years. Correlation analysis of IVOE (**A**), viscosity (**B**), appearance (**C**), taste (**D**), hardness (**E**), and fragrance for the two years, 2020 and 2021 (**F**). ** Indicates statistical significance at *p* < 0.01.

**Figure 3 foods-12-03267-f003:**
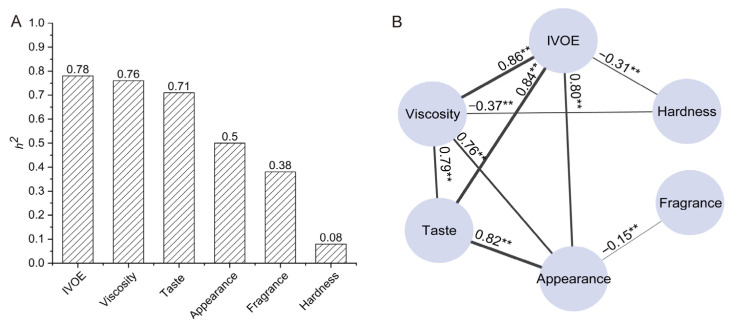
Broad-sense heritability of IVOE and its component traits (**A**) and correlation analysis of IVOE and its component traits (**B**). ** Indicates significance at *p* < 0.01.

**Figure 4 foods-12-03267-f004:**
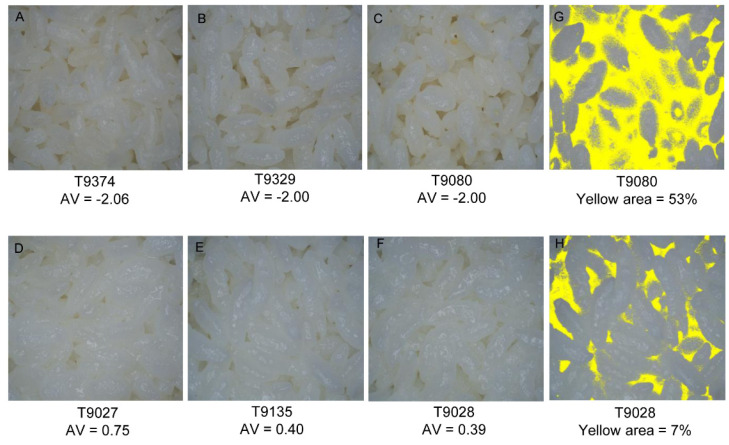
The yellow areas in cooked rice samples accurately reflect the appearance values. Cooked rice samples with different appearance values as determined by sensory tests (**A**–**F**), the extracted yellow areas for cooked rice samples with low (**G**) and high (**H**) appearance values. AV indicates the appearance value.

**Figure 5 foods-12-03267-f005:**
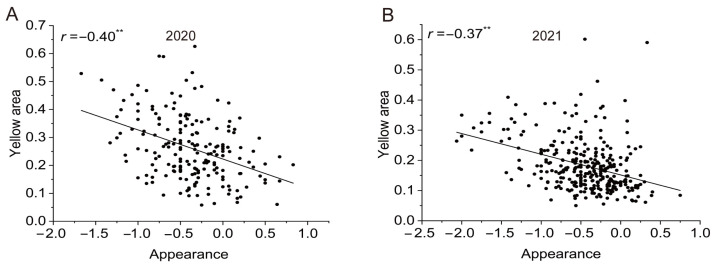
Correlation analysis of appearance values and image yellow areas for the two study years (2020 and 2021). Correlation analyses of appearance values and yellow areas in 2020 (**A**) and 2021 (**B**). ** Indicates significance at *p* < 0.01.

**Figure 6 foods-12-03267-f006:**
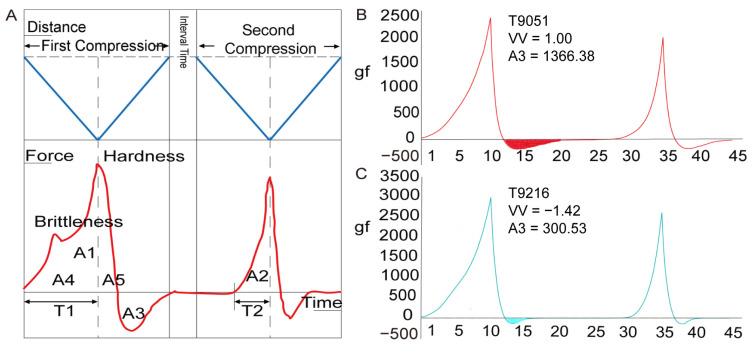
TPA parameters of cooked rice samples with different viscosity values. The texture analyzer measurement mode was texture profile analysis (TPA). Hardness: the peak power in the first downforce section; brittleness: smaller peaks before hardness; adhesiveness: the area of A3; springiness: T2/T1; chewiness: gumminess × springiness = A2/A1 × hardness × springiness; gumminess: A2/A1 × hardness; cohesiveness: A2/A1; resilience: A5/A4 (**A**). Texture profile of cooked rice with a high viscosity value (**B**). Texture profile of cooked rice with a low viscosity value (**C**). VV is the viscosity value.

**Figure 7 foods-12-03267-f007:**
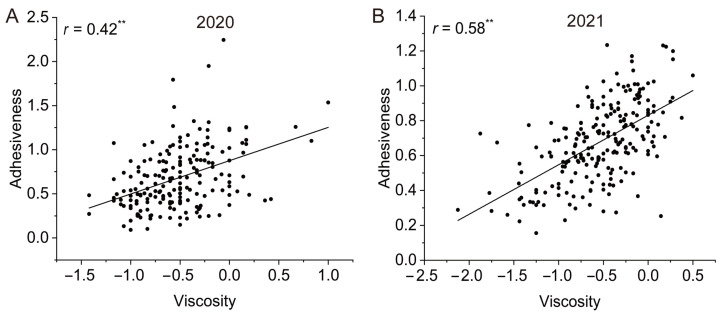
Correlation analysis of viscosity values and adhesiveness in the two study years (2020 and 2021). Correlation analysis of viscosity values and adhesiveness in 2020 (**A**) and 2021 (**B**). ** Indicates significance at *p* < 0.01.

**Figure 8 foods-12-03267-f008:**
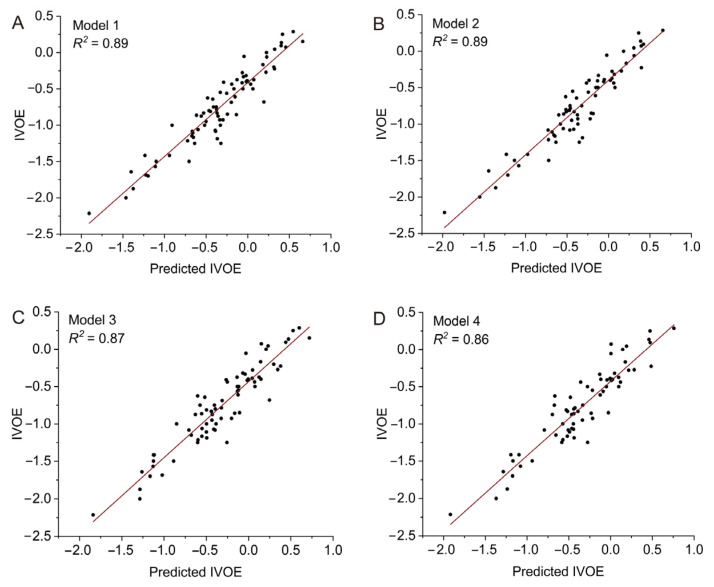
Correlation analysis of the predicted IVOE and the values of IVOE determined by sensory tests. Correlations between the predicted IVOE and sensory test IVOE for Models 1–4, respectively (**A**–**D**).

**Table 1 foods-12-03267-t001:** ANOVA of IVOE and its component traits.

Traits	Variation	DF	SS	MS	*F*	P
IVOE	Year	1	8.71	8.71	86.20 **	<0.01
	Variety	321	50.03	0.3	2.96 **	<0.01
	Error	321	16.87	0.1		
	Total variation	643	75.62			
Appearance	Year	1	2.06	2.06	18.84 **	<0.01
	Variety	321	29.33	0.18	1.60 **	<0.01
	Error	321	18.28	0.11		
	Total variation	643	49.66			
Viscosity	Year	1	5.24	5.24	77.34 **	<0.01
	Variety	321	29.43	0.18	2.60 **	<0.01
	Error	321	11.32	0.07		
	Total variation	643	45.99			
Taste	Year	1	3.49	3.49	37.03 **	<0.01
	Variety	321	33.12	0.2	2.11 **	<0.01
	Error	321	15.73	0.09		
	Total variation	643	52.33			
Hardness	Year	1	7.03	7.03	63.94 **	<0.01
	Variety	321	19.75	0.12	1.08	>0.05
	Error	321	18.36	0.11		
	Total variation	643	45.14			
Fragrance	Year	1	3.29	3.29	55.93 **	<0.01
	Variety	321	15.59	0.09	1.59 **	<0.01
	Error	321	9.82	0.06		
	Total variation	643	28.7			

DF: degrees of freedom, SS: sum of square, MS: mean square, F: *F*-statistic, P: *p*-value. ** indicates significance at *p* < 0.01.

## Data Availability

The data that support the findings of this study are available within the manuscript.

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
