# Peer review of "Evaluation of Cooked Rice for Eating Quality and Its Components in Geng Rice"

_foods, 2023, doi:10.3390/foods12173267_

Round 1
Reviewer 1 Report
1. Authors should write the keywords in alphabetical order
2. L. No. 22; change the format of the name of Japonica
3. Why font size is different for the introduction and other sections?
4. What genotypes authors used for this experiment? Authors should include those details in the materials and methods section. How many cultivars authors used for this experiment?
5. L. No. 124; Figure S1 or Figure 1? Why did the authors submit supplemental figures in the main manuscript?
6. How did the authors identify eleven well-trained panel members? Panel members were selected from the same region/institution or different regions/institutions? Did men and women include in the panel list?
7. Authors should include the environmental condition of the Experimental Station of Tonghua Academy of Agricultural Sciences in the materials and methods section.
8. What did the authors mean by control? Did the authors mean un-boiled rice?
9. Did the authors get any ethical committee for approval for this experiment? If yes, authors should submit an ethical committee approval number.
10. L. Nos. 117 and 193; check the font format of the acquisition
11. What is the difference between Figure S1 and Figure S2?
12. L. No. 131; how much volume of rice is used?
13. L. No. 145; write the full form of RGB.
14. L. Nos 151 and 152; remove the parenthesis.
15. Authors can reduce the materials and methods section and improve the results and discussion sections.
16. Did the authors use 322 japonica rice varieties for this experiment? If yes, the authors should include the accession number of each genotype. Where did the authors collect all these genotypes? Which genotype cooked quickly compared to the others?
17. Without checking Physiochemical properties (lipids, protein, carbohydrate, and more) how did the authors check the quality of cooked rice?
18. What are the factors affecting the quality of cooked rice during the analysis?
19. I recommended the authors should read Ken'ichi Ohtsubo and Sumiko Nakamura (https://www.intechopen.com/chapters/53218) and Yanjie et al 2018 (https://doi.org/10.1016/j.rsci.2018.10.003) articles. Both manuscripts will help the authors to improve their manuscripts.
Nil
Reviewer 2 Report
In general, the manuscript presents a comprehensive study on the evaluation of rice eating quality, focusing on the major component traits and their correlation with the overall eating quality index (IVOE). The authors have investigated various parameters such as appearance, viscosity, and taste and have attempted to establish objective indicators for their evaluation. The study makes an effort to bridge the gap between sensory evaluations and objective measurements, which is an important aspect of rice quality assessment. However, there are several issues that need to be addressed :
Rows 178-182: The text discusses the comparison of traits between two consecutive years but does not explicitly state the purpose of this comparison. Providing a brief rationale for why such a comparison is relevant would add clarity.
Rows 183-188: It would be beneficial to include a brief explanation of what "broad-sense heritability" refers to for readers who might not be familiar with the term.
In row 190: In subsection 3.1, the text talks about constructing an image acquisition system and using it to acquire images. However, it would be helpful to provide some technical details about how the image acquisition system was constructed and how the images were captured.
In row 214: "However, no parameters of the color models were correlated with cooked rice appearance" - This statement seems contradictory to existing color research. In fields like food science and color analysis, certain color parameters, such as L*, a*, and b*, are often used to describe appearance and are correlated with sensory perception.
In row 303: "Based on a collection of 322 japonica rice varieties that differed in eating quality" - While this statement suggests that the varieties were intentionally selected to represent varying eating qualities, it would be helpful to briefly mention how the variations in eating quality were determined or assessed.
In row 307: The text mentions "The values were in the ranges -2.50-0.40, -2.12-1.00, -2.06-0.49, -2.56-0.50, -0.93-1.12, and -1.31-0.58 for IVOE, viscosity, appearance, taste, hardness, and fragrance, respectively." - It's unclear what the ranges represent without further context. Are these score ranges for each trait, or some other form of measurement? Clarification is needed.
In the raw 340: "the correlation coefficients were 0.86, 0.84, and 0.80, respectively." - It would be helpful to specify which traits are being referred to in relation to these correlation coefficients.
In row 353: "Rice is mostly consumed in the form of whole grains, rather than after being processed into flour [12]." - While this statement is generally true, it's important to note that rice can also be consumed in various forms such as rice flour, rice noodles, and rice-based products like rice cakes
In row 355: "A number of QTLs for appearance, viscosity, taste, fragrance, and IVOE of cooked rice have been detected in the past two decades [25–31]." - This statement could benefit from briefly explaining what QTLs (Quantitative Trait Loci) are and how they are relevant to the study.
Please note that these are potential areas of concern or ambiguity based on the provided text.
In row 368: "The color rendering index of LED light is >90" - This statement lacks context. The color rendering index indicates how accurately a light source reproduces colors compared to a reference light source. While a higher value is generally better, stating ">90" without explaining the reference point or how it impacts the study could be confusing.
In row 370: The color temperature is 6500K, which is similar to afternoon daylight" - While 6500K is considered "daylight white" in terms of color temperature, stating that it is "similar to afternoon daylight" might be misleading, as natural daylight color temperatures can vary throughout the day and across locations.
In row 430: "will help to build a rice taste instrument suitable for japonica rice in northeastern China" - This statement lacks clarity on what a "rice taste instrument" entails and how it would be developed. It might be necessary to elaborate on the potential features, functions, and implications of such an instrument.
Considering the aforementioned points, I recommend that the authors perform a minor revision of the manuscript. Addressing the scientific inaccuracies, providing clearer explanations of methodologies, and further emphasizing the novelty and contribution of the research would significantly improve the manuscript's quality.
Round 2
Reviewer 1 Report
The authors addressed all the comments properly as per my suggestions. However, some mine correction is required to improve the manuscript. For example, authors can improve the legends of each main figure (not supplementary figure). Furthermore, the authors check the alignment of each main figure in the manuscript before publishing it.
Nil